# Using Inhibitory DREADDs to Silence LC Neurons in Monkeys

**DOI:** 10.3390/brainsci12020206

**Published:** 2022-01-31

**Authors:** Pauline Perez, Estelle Chavret-Reculon, Philippe Ravassard, Sebastien Bouret

**Affiliations:** 1Motivation, Brain and Behavior Team, Institut du Cerveau (ICM), INSERM UMRS 1127, CNRS UMR 7225, Pitié-Salpêtrière Hospital, 75013 Paris, France; pauline.perez2@icm-institute.org; 2Phenoparc Core Facility, Institut du Cerveau (ICM), INSERM UMRS 1127, CNRS UMR 7225, Pitié-Salpêtrière Hospital, 75013 Paris, France; e.chavretreculon@icm-institute.org (E.C.-R.); philippe.ravassard@icm-institute.org (P.R.); 3IVector Core Facility, Institut du Cerveau (ICM), INSERM UMRS 1127, CNRS UMR 7225, Pitié-Salpêtrière Hospital, 75013 Paris, France

**Keywords:** noradrenaline, chemogenetics, neuronal inhibition, locus coeruleus, primates

## Abstract

Understanding the role of the noradrenergic nucleus locus coeruleus (LC) in cognition and behavior is critical: It is involved in several key behavioral functions such as stress and vigilance, as well as in cognitive processes such as attention and decision making. In recent years, the development of viral tools has provided a clear insight into numerous aspects of brain functions in rodents. However, given the specificity of primate brains and the key benefit of monkey research for translational applications, developing viral tools to study the LC in monkeys is essential for understanding its function and exploring potential clinical strategies. Here, we describe a pharmacogenetics approach that allows to selectively and reversibly inactivate LC neurons using Designer Receptors Exclusively Activated by Designer Drugs (DREADD). We show that the expression of the hM4Di DREADD can be restricted to noradrenergic LC neurons and that the amount of LC inhibition can be adjusted by adapting the dose of the specific DREADD activator deschloroclozapine (DCZ). Indeed, even if high doses (>0.3 mg/kg) induce a massive inhibition of LC neurons and a clear decrease in vigilance, smaller doses (<0.3 mg/kg) induce a more moderate decrease in LC activity, but it does not affect vigilance, which is more compatible with an assessment of subtle cognitive functions such as decision making and attention.

## 1. Introduction

The noradrenergic nucleus locus coeruleus (LC) plays a critical role in various behavioral and cognitive processes including vigilance, arousal, attention and decision making [1,2,3]. Even if single unit recording in behaving animals provided a critical insight into the role of the LC, key advances have been made using manipulations of the noradrenergic system. Indeed, reversible manipulations of LC activity and/or noradrenaline (NA) receptors occupation in target regions clearly provided some insight about the causal relation between LC/NA activity and functions such as vigilance, memory or attention [4,5,6,7,8]. However, the pharmacology of the NA system is particularly challenging because NA acts on both alpha and beta receptors that show very distinct expression across the numerous targets of LC neurons [4,9]. Finally, pharmacology is inherently complicated by a set of technical issues such as specificity, the dynamics of action and the clearance or even accessibility to the target site. Thus, even if understanding the specific action of NA on distinct receptors located across several brain regions can provide a clear insight into the mechanisms underlying LC functions, the global influence of LC activity remains difficult to extrapolate from a collection of diverse local effects. Thus, to capture the function of the NA system, it is critical to understand the consequence of the global increase or decrease in LC activity, without any bias in terms of structure or receptor sub-type.

The recent development of pharmacogenetics has allowed key advances in understanding neuromodulatory functions [10,11,12]. Indeed, pharmacogenetics allow researchers to reversibly and reliably manipulate the activity of entire populations of neurons, with a high level of specificity [13]. The development of this approach in monkeys is critical for several reasons: First, it allows us to study behavioral and cognitive processes (and corresponding brain regions) that relay upon brain regions that are specific to primates [14,15]. Second, developing these approaches in primates is critical for translational research since monkeys are relatively close to humans from a phylogenetic perspective [16,17,18,19].

In recent years, we have developed a method to reliably and reversibly inactivate LC neurons using Designer Receptors Exclusively Activated by Designer Drugs (DREADD). The present paper describes the corresponding approach, as well as a set of preliminary data suggesting that the method could be used to study the role of LC neurons in cognitive operations in monkeys. We used a Lentivirus expression vector carrying a standard inhibitory DREADD (hM4Di). The vector was injected locally, but to ensure a specific expression in the LC, given its small size, we used a Tyrosine Hydroxylase (TH) promoter sequence such that only catecholaminergic neurons, but not neighboring cells, expressed the DREADD. After verifying the anatomical specificity of the DREADD expression in a first animal, we repeated the procedure in two other animals, and in one of them, we examined the influence of DREADD activation (with DCZ) on behavior and LC activity using single unit recording. As expected, we observed a reversible, dose-dependent effect of DCZ, with a clear decrease in vigilance and LC activity at maximal doses. Critically, intermediate doses of DCZ induced a more limited decrease in LC activity, but it did not affect vigilance. Thus, this method allows a reliable, reversible and dose-dependent manipulation of LC activity. The ranges of effects are clearly compatible with both studies on behavioral processes such as stress and vigilance but also on more specific cognitive operations, with doses of DCZ that do not affect vigilance.

## 2. Materials and Methods

### 2.1. Animals

All procedures complied with the guidelines of the European Community for the care and use of laboratory animals, applying the principle of 3Rs and in accordance with the welfare of the animals. According to European Community regulations (European Union Directive 2010/63/UE) and approved by the local Darwin Ethical and National committees (authorization #14611-2017122009291145 v4, 18/01/2019), experiments were performed on 3 rhesus monkeys (*Macaca mulatta*): A—male, 13 years old (9,5 kg); JF—male, 7 years old (13 kg); and JS—female, 7 years old (8 kg). They were hosted in the ICM primate facility.

Each monkey was trained by positive reinforcement to voluntarily sit in a primate chair to access the experimental box where the animals are trained to have their heads fixed in order to perform the electrophysiological recordings and care associated with the cranial implantations. They were positioned in front of a computer monitor on which we displayed various videos. After several weeks of habituation, monkeys could thus be comfortable for the entire duration of the experiment (several hours).

### 2.2. Viral Construct

Our general strategy was to use a lentivirus to carry the hM4Di DREADD gene. Given the shape of the LC (several millimeters in the AP axis but less than one mm in the ML axis), it would be very challenging to restrict the injection to the LC nucleus and cover a high enough proportion of LC neurons. Thus, we included a TH promoter sequence to restrict the expression of the DREADD to catecholaminergic neurons. We could inject a large enough volume to cover the entire AP axis of the nucleus with only 1–2 injections, and yet restrict the expression to the LC, which is the only noradrenergic nucleus in that area. Finally, to facilitate the localization of the material on brain sections, the construct included a fusion between hM4Di and a reporter protein (mCherry). A pilot experiment was conducted using material generously provided by B. Richmond and collaborators [20,21]. That construct also consisted in a lentiviral vector carrying the Gi-coupled hM4Di DREADD sequence under the control of a TH promoter, but CFP was used as a reporter gene.

#### 2.2.1. Lentiviral Vector Construct

Our objective was to express the Gi-coupled hM4Di DREADD in TH-positive neurons of the Locus Coeruleus using integrative gene transfer mediated by lentiviral vector. We used a fusion construct of hM4D(Gi) and mCherry that allows the coexpression of DREADD and the mCherry reporter gene under the control of the Cytomegalovirus ubiquitous promoter (pCMV) in a third-generation lentiviral backbone (pLenti6.3, Life Technologies, Carlsbad, CA, USA). The pLenti6.3/V5-pCMV-HA-hM4D(Gi)-mCherry construct (kindly provided by the iVector platform) was digested with BspDI and SalI to remove the pCMV promoter. We used a 2362 pb long fragment of the rat TH promoter to drive expression of the hM4Di transgene. This fragment corresponds to the proximal TH promoter that includes the transcription start site. Such area of the promoter shows strong homology with Macaque and human promoters (>80%). In addition, based on P. Ravassard’s unpublished data, we have observed that such a fragment is able to restrict the expression of a reporter gene in human-iPS-derived Dopaminergic and Noradrenergic cells. The 2362 pb fragment of the Rat TH promoter was amplified by PCR from a 5.3 kbpTH-GFP vector provided by P. Ravassard [22] using the following Gibson Assembly (lowercase) compatible primers: Forward 5′ tacaaaaattcaaaattttatcgatGGCCTAAGAGGCCTCTTGGG 3′ and reverse 5′ gtgaagttggccatggtggcgtcgacTGTACCCAGTGCAAGCTGGT 3′. The resulting PCR product was next inserted into the BspDI-SalI linearized lentiviral vector using Gibson Assembly kit (NEB #E2611S) according to manufacturer instructions to produce the pLenti6.3/V5-pTH-HA-hM4D(Gi)-mCherry final lentiviral construct

#### 2.2.2. Lentiviral Vector Production

The pLenti6.3/V5-pTH-HA-hM4D(Gi)-mCherry lentiviral vector was produced by co-transfection in HEK-293T cells along with the following plasmids pRSV-REV expressing REV, pMDLg/pRRE expressing gag and pol and pHCMV-G encoding the VSV glycoprotein-G, as described previously [23]. The supernatants were treated with DNAse I (Roche Diagnostics, Basel, Switzerland) prior to their ultracentrifugation, and the resultant pellets were re-suspended in PBS, aliquoted, then frozen at −80 °C until use. Titer was determined using real time quantitative qPCR as described previously [24] and corresponded to 8.10^9^ transduction unit per milliliter (TU/mL). The construct was stored frozen at −80 °C in aliquots of 60 microliter.

### 2.3. Surgery

Given the small size of the LC and its location, deep into the brainstem, it was critical to be able to locate it very accurately and calculate a trajectory that avoids both ventricles and major blood vessels. In addition, since the LC is not visible on MRI scans and since we wanted to be able to verify its position using neurophysiology, we adapted the approach used for single unit recordings [25]. In short, we placed a recording cylinder on the skull above the LC in each hemisphere and used a stereotaxic grid to position the injection devices reliably. The location of the grid and of the injection devices was verified using a combination of anatomical MRI and neurophysiology.

#### 2.3.1. Head Post and Recording Cylinders

First, we scanned the monkey using a 3T MRI scanner (Siemens, Erlangen, Germany) to determine the position of the LC and calculate a safe trajectory for the injection devices (Figure 1A). Note that even if the LC was not visible on our MRI scans, we could easily identify its position on each animal using standard landmarks and brain atlases. Once the position of the LC was identified in each hemisphere, we determined the position of the recording cylinder on the skull such that the angle and the position allowed us to reach the LC without puncturing lateral ventricles or major blood vessels such as the posterior cerebral artery.

Then, we implanted the 2 recording cylinders (one on each hemisphere) as well as a head post on the skull of the animal by performing a sterile surgery under general anesthesia in a fully equipped surgical room. As classically performed for neurophysiology in monkeys, the recording cylinders were positioned on the skull at the intended stereotaxic coordinates (based on the MRI scans), and they were held in place using dental cement secured on the skull via platinum screws. The head-post (to hold the animal’s head), was also held in place via dental cement. Upon recovery from surgery, the monkey was progressively habituated to be head-fixed, such that it could tolerate recording sessions of about 1 to 2 h. After about 4 weeks of recovery after implantation, we opened the skull in each of the recording cylinders to allow access to the brain. We scanned the monkey again with a marked (iodin) stereotaxic grid (Crist Instrument) to identify which position should be used for accessing the LC in that grid (Figure 1B). Note that this could be performed before or after opening the skull. Subsequently, we verified the position of the LC in the coordinate of the stereotaxic grid using MRI and/or neurophysiology. To identify the injection site within the stereotaxic reference of the grid, we placed a tungsten recording electrode at a given depth in one of the grid holes, and the animal is scanned with the electrode in place. The artefact of the electrode is easily visible on T1 images (Figure 1C).

#### 2.3.2. Electrophysiology

We used neurophysiological recordings to confirm the position of the LC, using standard criteria [25,26,27,28]. These criteria include relatively broad spike shapes >660 μs, filtered), slow and irregular firing rates (2–3 spikes per seconds) and biphasic (activation followed by inhibition) response to salient stimuli. Their firing rate is also closely related to vigilance, with a marked decrease in firing when the animal becomes drowsy. Finally, the LC is located just ventral to the MesV nucleus, which controls the face and thus responds to facial movement and stimulation. Electrophysiological recordings were made with tungsten microelectrodes (125 mm diameter, FHC), which were positioned in the stereotaxic grid, introduced into the brain through the dura mater with a guide tube and lowered to the theoretical position of the LC with a micro-manipulator (Narishige, Tokyo, Japan). The neurophysiological signal was amplified and band-passed filtered (0.3–6 kHz, OmniPlex, Plexon, Dallas, TX, USA).

Once the position of the LC was identified, we calculated the coordinates of the injection sites such that injections could be made 1 mm above the LC. We identified two injection sites per hemisphere, one on the anterior part and one on the posterior part of the nucleus.

#### 2.3.3. Injection

The surgery took place in a BSL2 sterile surgical suite, under aseptic conditions and general anesthesia. Physiological constants were monitored continuously, and the animal received appropriate analgesia and antibiotic treatments. The injecting device (Figure 2A) was constructed using a 31G stainless steel tube (injector) attached to a 0.01” tubing (Tygon PVC Tubing, 0.010” ID × 0.030” OD), sealed with neoprene glue. The end of the tubing was then attached to a 27G needle mounted on a 100 µL Hamilton syringe (Gastight TLL, #1710). A small point of indelible marker was made on the injector to identify the depth at which we needed to descend it, relative to the guide tube.

For one animal (monkey A), only 1 hemisphere was injected during the initial surgery, and the second hemisphere was injected 9 days later. For the other 2 monkeys (JF and JS), both hemispheres were injected during the same surgical procedure.

The monkey was placed in a stereotaxic frame, and the grids were inserted in the chambers in the exact same position as during the MRI scanning and electrophysiological recordings. A 25G guide-tube and its stylet were introduced at the position previously identified with MRI images. The syringe pump and the needle holder were both held by different micromanipulators, which were mounted onto the stereotaxic frame (Figure 2B). The syringe was filled with mineral oil (Advanced Air Tool Oil, Sip Industrial, Loughborough, UK) with all precautions taken to avoid gas intrusion. Then, the 27G Luer needle was attached to the syringe, and the plunger was pushed until it reached 40µL to fill it with oil and expel the air inside the needle. The viruses were withdrawn by the syringe under a Biological Safety Cabinet at a rate of 20 µL/min before connecting the tubing to the needle. Then, the content was pushed inside the injecting device at a rate of 5 to 10 µL/min until it reached the tip of the injector. At this point, the injection speed was slowed down to 0.1 µL/min and the injector was inserted inside the brain through the guide-tube and slowly moved down 1 mm above the target. A 2 min waiting period was allotted before starting the 10µL injection, at a rate of 0.5 µL/min. The injector was kept stationary for 10 min after the injection to allow for stable diffusion of the solution into the LC.

### 2.4. Histology

Monkey A was euthanized 63 days after first injection, 54 after second injection. The animal was first anesthetized with Ketamine and then euthanized (Euthazol, IV, 100 mg/kg) under anesthesia with constant monitoring. Right after the animal’s death, it was perfused, first with a solution of saline/heparin for total exsanguination, then with paraformaldehyde (4%) for tissue fixation. After extraction, the brain was cryoprotected by successive immersion in increasing concentrations of sucrose (10%, 20% and 30%). It was then frozen in dry ice and sliced using a microtome into 40 µm sections. The floating sections were then kept in azide solution at +4 °C.

#### 2.4.1. Direct Fluorescence

Immediately after sections were obtained, some of them were mounted on glass slides and examined under a fluorescence microscope. We focused on the LC region searched for the fluorescence emitted by the fluorescent protein (CFP) coupled to the hM4Di (see above ‘viral construct’). We used a green filter because no blue filter was available. As shown in Figure 3, a clear fluorescent signal was visible near the fourth ventricle at the expected location of the LC neurons.

#### 2.4.2. Immuno-Histochemistry

We used immuno-histochemistry to detect the expression of the hM4Di DREADD and to compare it with the expression of TH, which characterizes LC neurons. All procedures were conducted at room temperature, and all steps were performed under agitation. On day 1, brain sections were rinsed 3 times in PBS for 8 min/wash. The sections were then incubated for 15 min in a solution containing 20% methanol +3% H_2_O_2_ in PBS to inhibit endogenous peroxidase activity and rinsed 2 times in PBS for 8 min. They were then transferred in a 0.2% triton solution in PBS for 15 min to allow tissue permeabilization and washed twice in PBS for 8 min. To limit non-specific binding, sections were incubated for 30 min in 10% donkey serum +10% BSA in PBS and rinsed twice in PBS (8 min). Sections were then incubated with the primary antibodies diluted in PBS for 24 h at 4 °C. We used a mouse anti-TH antibody (Immunostar, ref 2294, 1/1000) to label TH and a rabbit anti-GFP antibody (Abcam, ref ab290) 1/750) to detect the CFP coupled with the hM4Di DREADD (see above ‘viral construct’, [20]). On day 2, sections were rinsed 2 times for 8 min in PBS before incubation with secondary antibodies. From that point, the procedure was conducted in a dark room. The sections were incubated for 1 h either in donkey anti-mouse IgG (dilution in PBS: 1/1000; coupled with Alexa 555 (orange fluorescence) Invitrogen, ref A21202) to detect TH or in donkey anti-rabbit IgG (dilution: 1/1000; coupled with Alexa 488 (green fluorescence); Invitrogen, ref A31572) to reveal the HM4Di-CFP DREADD. They were then rinsed 2–4 times for 8 min/wash in PBS and mounted on gelatin slides.

Mounted sections were examined under a fluorescent microscope (Axioskop 2 plus, Leica). The first observations of TH immuno-labelled sections revealed a relatively high level of green fluorescence background (Figure 4A). Even if the signal was clearly stronger in the LC compared to the background, we decided to perform immunohistochemistry with diaminobenzidine (DAB) to confirm the specificity of DREADD expression in LC in another set of sections. Sections were pretreated with 20% methanol + 3%H2O2, 0.2% Triton X-100 and 10% normal goat serum + 10% Bovin Serum Albumin and then incubated in the antibody solution (IGg, anti-GFP, Abcam, ab 290, 1/750) for 24 h. This was followed by incubation in secondary biotinylated antibodies (1/125), then in avidin–biotin–peroxidase complex (ABC) solution, 0.01 M DAB (1/1) and 0.008% H_2_O_2_ for 10 min. Sections were mounted on gelatin slides and sequentially dehydrated in ethanol solutions (70, 80, 90, 100%) for 2 min. Then, the sections were incubated in xylene 3 times for 2 min each. As shown in Figure 4B, immuno-histochemistry confirmed the strong and specific expression of the HM4Di-CFP DREADD in the LC. Finally, as shown in Figure 5 the expression of the fusion protein hM4Di-CFP strongly overlapped with the expression of TH, which confirms that a majority of LC neurons expressed the hM4Di DREADD.

### 2.5. Functional Validation: Effect of DCZ Injection

We used deschloroclozapine (DCZ) as a selective agonist of the DREADD receptor, based on recent work by Nagai et al. (2020). DCZ was dissolved in a saline-DMSO 2% solution and injected intramuscularly. We first injected one animal (monkey JF) in its home cage (0.1 mg/kg) and observed it constantly for at least one hour. We evaluated changes in vigilance using both locomotor activity and posture (standing vs. laying down), as well as eye opening. We did not observe any clear change in behavior, nor any decrease in vigilance or any motor or attentional deficit.

Subsequently, we evaluated the joint influence of several doses of DCZ on behavior and LC activity. We also used monkey JF, which was very familiar with the recording setup and the single-unit recording procedures. The monkey was sitting quietly in a primate chair facing a monitor on which we displayed various videos while we lowered the electrode into the brain. The electrode was advanced slowly until we identified LC units based on standard criteria (see above). Once we had isolated at least one LC unit, we waited for 5–10 min before injecting the DCZ (intramuscular injection). The animal was monitored constantly during the course of the experiment. We completed a total of 5 successful experiments and assessed the effect of 4 doses of DCZ on 9 LC units: 0.1 mg/kg (*n* = 2 sessions, 5 LC units); 0.2 mg/kg (*n* = 1 session, 1 LC unit); 0.4 mg/kg (*n* = 1 sessions, 1 LC unit); 0.5 mg/kg (*n* = 1 session, 2 LC units). In all of these experiments, the activity of LC units was followed for at least one hour.

Behaviorally, as it was the case in the home cage, neither of the 2 injections of DCZ at 0.1 mg/kg had any apparent effect on the behavior. The dose of 0.2 mg/kg (*n* = 1) did not have any clear effect either. Although we cannot rule out a moderate effect on vigilance, the animal did not close its eyes for more than a few seconds in a row. By contrast, at 0.4 and 0.5 mg/kg (*n* = 1 for each dose), the animal showed clear signs of decreased vigilance (decreased locomotor activity and clear increase in the amount of time spent with eyes closed). These behavioral effects appeared within 5–10 min and lasted at least 1 h after injection.

The influence of DCZ on LC activity also varied according to the dose, even if the sample was clearly too limited to draw any strong conclusion. We measured the firing of 9 LC units for at least 2 h around the injection of DCZ. For each unit, we computed the firing rate in bins of 10 s and compared the mean spike count per bin between the 10 min period preceding the injection and a window of the same size but starting 2 min after the injection. A summary of the analysis is provided in Table 1. We considered that a neuron responded to the drug when its firing differed significantly between the 2 periods (*t* test, *p* < 0.01). At 0.1 mg/kg, 3 of the 5 LC units displayed a clear decrease in activity (1/3 units in one session; 1; 2/2 in the other session). At 0.2 mg/kg, the one LC unit recorded did not show any change in firing rate following injection. By contrast, at 0.4 and 0.5 mg/kg, all LC units recorded (*n* = 1 and *n* = 2, respectively) showed a decrease in activity. At all doses, the influence of DCZ on firing appeared within 5 min and lasted for the entire duration of the experiment, i.e., at least 1 h after injection. The average time course of the firing around the injection is shown in Figure 6. To further quantify the magnitude of the firing rate change after DCZ injection, we measured the percentage of firing rate change after injection (using the same windows as before). For doses smaller than 0.3 mg/kg (*n* = 6 neurons), the firing decreased to 88 ± 9% of baseline activity. For doses greater than 0.3 mg/kg (*n* = 3 neurons) m, the firing decreased to 40 ± 10% of baseline activity. Even with so few neurons, the difference between the two is significant (*t* test, *t*(5.8) = −3.6, *p* = 0.01). Finally, note that we also recorded 2 units that were clearly not LC units, and none of them changed their firing after DCZ injection (0.1 and 0.4 mg/kg).

In summary, for DCZ doses of 0.1 and 0.2 mg/kg, we observed no clear behavioral effect (no change in vigilance), but about half of the neurons (*n* = 3/6) showed a decrease in firing rate. For higher doses of DCZ (0.4 and 0.5 mg/kg), we observed a clear decrease in vigilance and all recorded LC units (*n* = 3/3) showed a decrease in firing.

## 3. Discussion

In summary, these data indicate that chemogenetics can be used to specifically and reversibly manipulate the activity of LC neurons in monkeys. In short, the complete injection procedure requires several steps, spanning over weeks, but it is reliable. The injection parameters allow for a complete coverage of the LC, and the expression of the DREADD is restricted to NA neurons. DREADD activation with DCZ seems to induce a dose-dependent, but partial, inactivation of LC neurons, which can lead to a reversible decrease in vigilance at the highest doses. Thus, chemogenetics can be used to reversibly manipulate LC activity in awake behaving monkeys performing standard laboratory tasks. It should provide a way to reliably and reversibly manipulate LC activity over extended periods of time and to explore the effects of these specific manipulations on both behavior and neuronal activity in monkeys.

In recent years, chemogenetics has become a very efficient method to reversibly manipulate large populations of neurons in monkeys [20,29,30,31,32]. Here, we demonstrate that the method can be used to target specific populations of neurons over a relatively long time (several hours), with an efficacy of inactivation falling within the expected range, i.e., sufficient to obtain functional effect but limited enough on vigilance to allow the monkeys to perform standard laboratory tasks.

We used a lentivirus vector to carry the HM4Di construct (combined with a fluorescent protein (to facilitate detection)) and the TH promoter, and we believe that this choice is appropriate for our current aims. First and foremost, the infection is both complete and specific, and the functional effects are within the expected range, so there is no major incentive to consider an alternative approach. Because lentiviral vectors are integrative vectors, they might be thought to induce mosaic expression if the expression levels across cells vary as a function of where exactly the vector is integrated in the genome. However, integration, although considered as random, is in fact preferentially observed in active regions of the chromatin [33]. In addition, transduced cells often contain multiple integration sites that usually normalize the expression level between transduced cells that are all different in terms of chromosomal integration. It is true that epigenetic silencing could contribute to reduce the long-term expression, and this is clearly observed when using, for example, a CMV promoter that is well-known to be sensitive to methylation [34]. When using cell-specific promoters, the situation is different, since such promoters are active in the target cell type and therefore will not be epigenetically extinguished. The integration site could be responsible for the epigenetic extinction of the exogeneous promoter, but this will occur in small number of transduced cells, and integration is different from one cell to the other.

Second, the expression of the DREADD should be stable over several years, which could enable us to run long experiments involving not only behavior but also neurophysiological recordings. Given that such experiments could take over a year, we feel that using lentivirus is perfectly adequate. This is also critical for translational purposes, since using such viral-vector-based methods in patients also requires the expression of the transgene to last over several years [18]. Third, the spread of the virus should be relatively limited to avoid reaching other brainstem catecholaminergic nuclei, and in that respect, the spread of lentivirus injection in our experiments was sufficient to cover the entire LC nucleus, yet it did not reach other nuclei. From that perspective also, the method presented appears perfectly adequate: For the one monkey in which we obtained histological sections, the expression of DREADDs could be seen with native fluorescence of the coupled CFP on fresh sections. Immuno-histochemistry revealed that the injection was sufficient to cover the entire LC, from its rostral to its caudal part. In addition, the overlap between TH and HM4Di-CFP DREADD expression was very good, suggesting that an apparent majority of LC cells expressed the construct. We did not verify the subcellular localization, but the functional effects of DCZ injection indicates that a sufficient amount of hM4Di receptors was present in the membrane of LC neurons to respond to DCZ. Lastly, even if the coverage of LC neurons seemed relatively homogenous, this remains difficult to assess given the anatomical heterogeneity of the nucleus in monkeys [13,35]. Even if a finer characterization of that heterogeneity would be critical to better understand the specific functions of LC in primates, we believe that the current method is sufficient to induce a reliable activation of LC neurons, and along with a finer anatomical characterization of the effects, further functional testing would also be critical to evaluate its efficacy.

The method used here for targeting the LC is very demanding compared to standard stereotaxic injections: In addition to identifying the target location in stereotaxic space using MRI, we place a recording cylinder and use a stereotaxic grid, which provide a very reliable positioning in the A/P and M/L axes (e.g., [25,36]). To confirm the accurate location of the injection in the D/V axis, we scan the animal after placing an electrode in the grid at a known depth. This method, inspired from the approach used for LC neurophysiology, has proven very reliable and enabled us to reach the LC with the required precision (mm range). Finally, the implant over the monkey’s head enabled us to conduct single unit recording and compare the influence of DCZ injection on behavior and LC activity. Given the relation between LC activity and vigilance, and given the difficulty to control the expression level of DREADDs in target neurons, it seems essential to validate the functional impact of DCZ injection by evaluating the firing of LC neurons.

We chose to use DCZ to activate the DREADDs. Indeed, DCZ was shown to have greater affinity with DREADD receptors while reducing binding with off-targets receptors such as serotoninergic or dopaminergic receptors, compared to other hM4Di ligands such as CNO, C21 and clozapine [31]. In that same study, the authors demonstrated that DCZ induced the rapid (less than 10 min) activation of the hM4Di and HM3Dq in both rats and monkeys. Here, the dose necessary to induce behavioral effects (0.4 mg/kg) was larger compared to previous studies (0.1 mg/kg). Of course, one simple account is that we are studying a different structure, and the effect on vigilance might be more difficult to capture, especially without finer measures such as EEG. However, on the other hand, even when DCZ had no apparent effect on behavior (at 0.1 or 0.2 mg/kg), three out of six LC neurons displayed a significant decrease in activity. At higher doses (0.4 or 0.5 mg/kg), DCZ induced a reliable decrease in vigilance, and all three LC neurons recorded during these experiments showed a decrease in firing. We acknowledge that our sample is very small, and it is very difficult to draw any strong conclusion regarding the relation between behavioral and neurophysiological effects of DCZ-induced HM4Di activation. Still, increasing the dose of DCZ seemed to increase the proportion of LC neurons inhibited, which is in line with the associated effect on vigilance, which was only visible at higher doses. Critically, hM4Di activation did not cause a complete inhibition of LC neurons, and both the animal and the neurons remained sensitive to alerting stimuli (e.g., a knock on the door). Thus, further studies would be necessary to characterize the functional relation between the impairment of LC activity and LC functions.

Interestingly, the dose–response effect of DCZ in these experiments is reminiscent of those of the alpha 2 agonist clonidine, which inhibits LC neurons by activating Gi coupled alpha-2 auto-receptors [37]. Indeed, clonidine induces both a clear decrease in vigilance and a reliable decrease in LC at 20 μg/kg, but importantly, we could observe reliable behavioral effects on decision making with smaller doses of clonidine (<10 μg/kg) that had virtually no effect on vigilance [36,38]. Thus, we believe that a range of DCZ concentrations described by Nagai et al. (2020), which are sufficient to specifically activate a large fraction of DREADDs, could produce a reliable functional perturbation of LC activity. Indeed, previous studies showed that behavioral effects could be readily observed with only 3% of cortical neurons infected [32], which suggests that the behavioral effect might be related to an impairment of network functions rather than a total inhibition of firing. We are very well aware that these results are very preliminary and that, in the absence of proper controls, care should be taken with the interpretation. In particular, we cannot rule out that the high doses of DCZ (0.5 mg/kg) induced non-specific effects by acting on endogenous receptors rather than on the DREADDs, as suggested by a recent study [39]. However, given the nature of the effects described here, the dose effect on neuronal activity and the similarity with clonidine (which decreases LC activity), we are relatively confident that the effects of DCZ observed here are related to its action on DREADDs expressed in the LC. Indeed, at doses where DCZ is known to be specific (<0.3 mg/kg, see [31,39]), LC neurons show a moderate decrease in activity (Figure 6) and the behavioral effects observed at higher doses could simply be explained by the stronger effects on LC neurons.

Even if it is technically more challenging, chemogenetic inactivation with DREADDs offers consequent advantages relative to pharmacological inactivation with clonidine. Indeed, it is obviously more specific, since alpha 2 receptors are present not only in the LC but also in the prefrontal cortex, where they exert a powerful effect on executive functions [4,37]. In addition, there is presumably less variability compared to pharmacology because it is independent of inter-individual difference in receptor expression and physiology (e.g., crossing the blood–brain barrier, clearance). Indeed, we did not observe any sign of habituation after repeated daily injections, and the contrast of effects across different doses used was very clear (0.1: not much on vigilance, 0.2: quiet, 0.5: sleepy). Finally, the duration of effects is much greater than those of clonidine (>2 h; vs. ∼1 h). As Nagai et al. (2020) reported, a single dose of DCZ (0.1 mg/kg) induces a reliable effect over more than 2 h. A similar timing was reported for CNO-mediated DREADD inhibition of the striatum [29]. Here, the effects of the drug were stable over the entire duration of the recording sessions (>1 h), but they had completely disappeared after 24 h, which would enable us to conduct daily sessions as usually performed with laboratory monkey experiments. Thus, the influence of DREADD-mediated LC inhibition should provide a significant methodological improvement over pharmacology.

In the future, a reliable and reversible inactivation of LC neurons will open the way for further studies aimed at evaluating how the role of the LC/ NA system in behavior and cognition is mediated by its action on its target structures. After seminal studies using pharmacology, combining specific and reversible inactivation of the LC could be combined with imaging methods to relate changes in behavior with changes in target networks [12,40,41]. Even if temporal resolution is not sufficient to study transient effects, this is a critical first step to evaluate the functional influence of LC activity on behavior and target networks.

In conclusion, this method provides a reliable way to selectively and reversibly inactivate LC neurons using chemogenetics in rhesus monkeys. It should prove very useful to study the role of the LC in behavioral and cognitive processes in primates. This should be very complementary to the work conducted in rodents using similar approaches. In the future, this method could easily be adapted to modulate LC activity with other DREADDs (activating ones) or optogenetics. It might also be adjusted to target other brainstem neuromodulatory nuclei.

## Figures and Tables

**Figure 1 brainsci-12-00206-f001:**
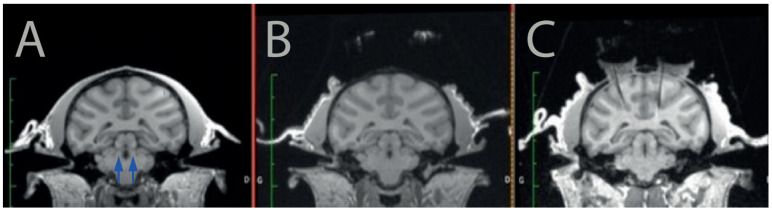
Targeting the LC with MRI. The monkey was first scanned before surgery (**A**) to identify potential trajectories toward the LC (arrows) and to reach it from the surface of the skull without piercing a ventricle or a blood vessel. After the recording cylinders and headpost were implanted surgically, we marked a stereotaxic grid with iodin and scanned the monkey with the grid in place (**B**). After the grid hole that would enable to reach the LC was identified on this second scanner, we scanned the monkey a third time with an electrode inserted in the corresponding grid hole (**C**). The artefact generated by the electrode is clearly visible on the image, and it shows that the trajectory is adequate to target the LC.

**Figure 2 brainsci-12-00206-f002:**
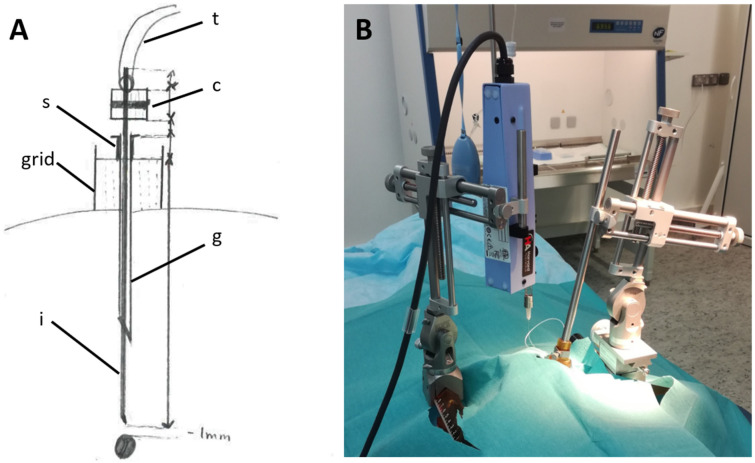
Injecting the viral construct. (**A**) Injecting system: First, the guide-tube (*g*) is inserted into one of the grid holes. A spacer (*s*) prevents the guide-tube from descending under the surface of the grid. Then, the injector (*i*) is lowered into the brain through the guide-tube until it reaches the target (1 mm above the LC). It is held in place by the needle holder and connected (*c*) to the flexible tubing (*t*). (**B**) During an injection: The two manipulators are mounted on the stereotaxic frame, one holding the Hamilton syringe (left), the other holding the injecting system (right), both connected to each other by the flexible tubing. Here, the injector is already in position in the brain.

**Figure 3 brainsci-12-00206-f003:**
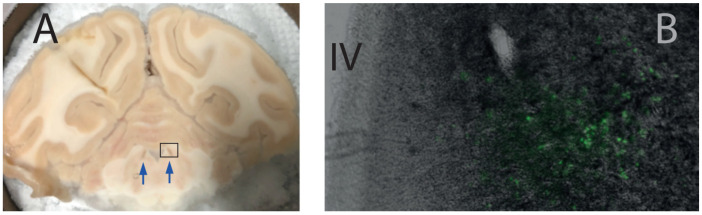
Histological validation of DREADD expression in the LC. Panel (**A**) shows a brain section at the level of the LC (brown nucleus, arrow), which is clearly visible on the side of the IVth ventricle. The rectangle on panel (**A**) shows the approximate location of the section on panel (**B**). (**B**) Photomicrograph of a 40 μm section taken with a fluorescence microscope. The fourth ventricle (IV) is visible on the left side of the picture, and the fluorescence emitted by the reporter protein is clearly visible at the expected location of the LC. Note that we used a green filter to capture the fluorescence of the CFP.

**Figure 4 brainsci-12-00206-f004:**
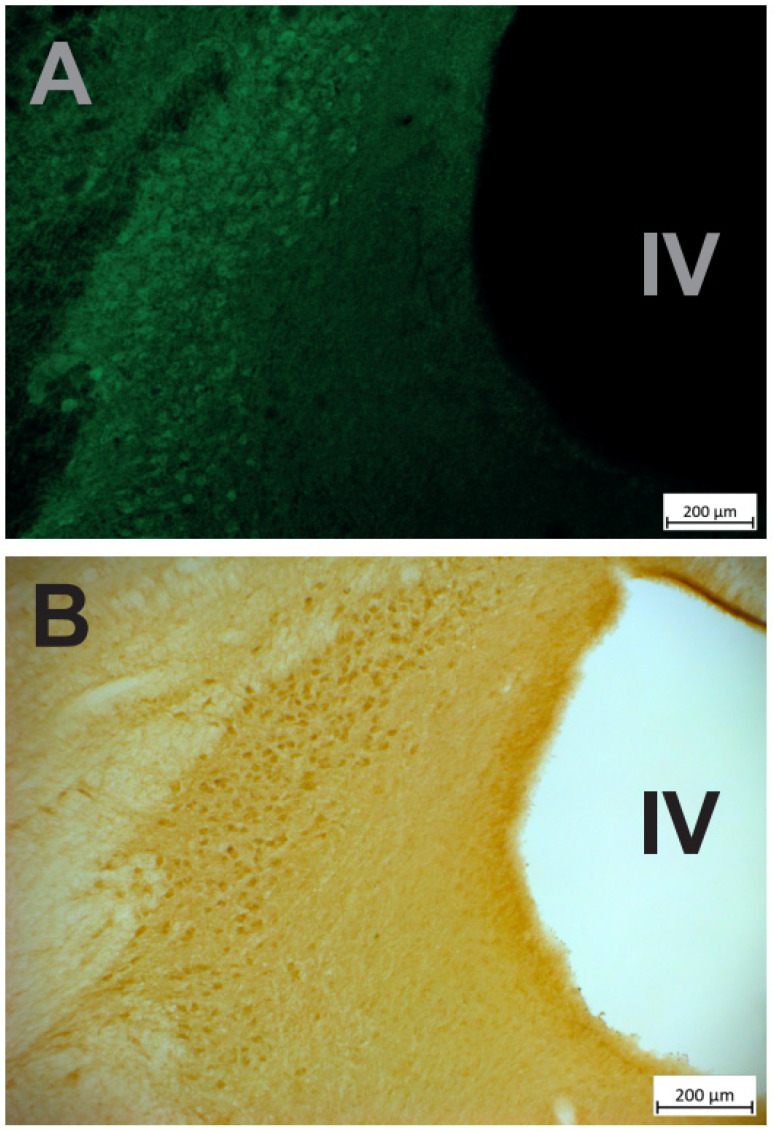
Immuno-detection of the DREADD. (**A**) Immuno-fluorescent labeling of the HM4Di-CFP DREADD, the LC is clearly visible, just lateral to the fourth ventricle (IV), in spite of the green background covering the entire section. (**B**) A neighboring section was treated with immuno-histochemistry to reveal the expression of the HM4Di-CFP DREADD. The LC is also clearly visible, with little staining in surrounding nuclei.

**Figure 5 brainsci-12-00206-f005:**
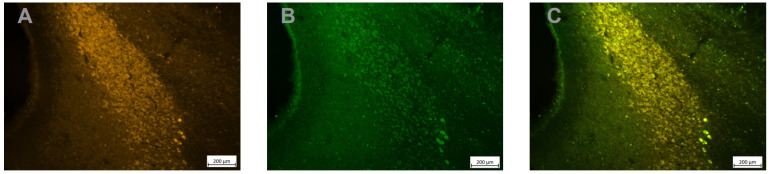
Strong DREADD expression in TH+ LC neurons. Adjacent sections treated with an anti-TH antibody (**A**), an anti GFP antibody to reveal the HM4Di-CFP DREADD (**B**) and an overlay of the 2 images (**C**) shows the perfect overlap between TH expression and DREADD expression in LC neurons.

**Figure 6 brainsci-12-00206-f006:**
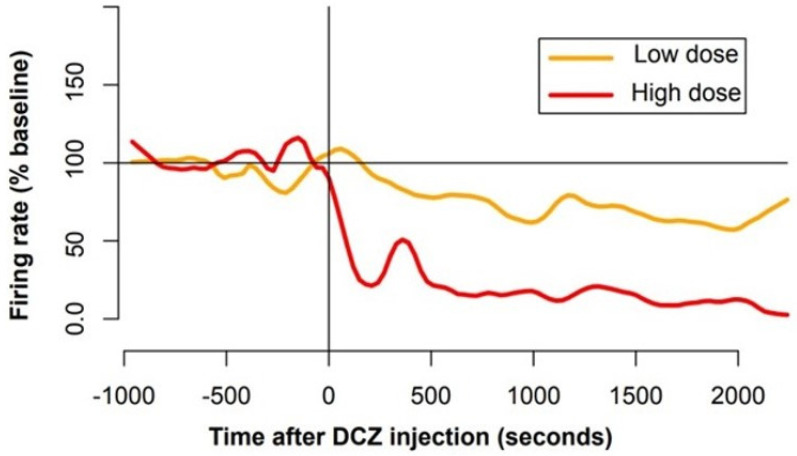
Influence of DREADD activation on LC activity. Averaged firing rate of LC units was computed separately for neurons tested with high doses of DCZ (in red, 0.4 mg/kg, *n* = 1 and 0.5 mg/kg, *n* = 2) and for neurons tested with a lower dose (in orange, 0.1 mg/kg, *n* = 5 and 0.2 mg/kg, *n* = 1). Individual firing rates were normalized using a 5 min baseline window, and data were smoothed for clarity. The injection (t = 0, vertical line) induced a rapid decrease in activity, which was more pronounced for higher doses of DCZ. The activity of LC neurons decreased rapidly, and after 15 min, they remained lower than baseline for at least 1 h.

**Table 1 brainsci-12-00206-t001:** Influence of DCZ injection on individual LC units.

Cell Nb	DCZ (mg/kg)	Firing Rate (Pre-Injection)	Firing Rate (Post-Injection)	% Firing after Injection	Significant Difference
1	0.1	4.8	3.3	69	yes
2	0.1	2.4	2.9	120	no
3	0.1	3.6	4.1	113	no
4	0.1	1.4	0.8	72	yes
5	0.1	1.5	1.0	67	yes
6	0.2	5.7	5.0	88	no
7	0.4	1.9	1.1	58	yes
8	0.5	7.4	2.7	36	yes
9	0.5	6.7	1.7	25	yes

## Data Availability

Not applicable.

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
