# Peer review of "Using Inhibitory DREADDs to Silence LC Neurons in Monkeys"

_brainsci, 2022, doi:10.3390/brainsci12020206_

Round 1
Reviewer 1 Report
In the present study, the authors describe a method to chemogenetically control the activity of the Locus Coeruleus (LC) in macaques. They use stereotaxic injection of Lentiviruses encoding hM4Di-mCherry or -CFP under the control of a fragment of the tyrosine hydroxylase (TH) promoter. The advantage of this promoter is that it should only be active in cells synthetizing catecholamines, as this requires the hydroxylation of tyrosine via TH. The authors used three monkeys in the study, A, JS, and JF into whose LC’s they injected variants of the hM4Di-Lentiviruses. They asses expression with immunohistochemistry and evaluate chemogenetic silencing of the LC by rating vigilance and by measuring single unit activity in the LC. In principle, the study is interesting and might provide some insights into better ways of controlling LC in macaques. However, there are a number of problems with the manuscript.
First, the authors make a number of claims for which they do not provide any or only minimal data. Most problematically, they rely heavily on the analysis of vigilance. However, there is not a single data point showing any quantification of this parameter. Moreover, a description how vigilance was measured or scored is missing. If quantifying vigilance proves difficult, one way to inspect the activity of LC in a more quantitative way could be to measure pupil dilation.
Next, the authors deduce the effects of different concentrations of deschloroclozapine (DCZ) on LC activity by recording single units. Also, in this case, almost no data is shown. There is one exemplary histogram showing firing of a single unit after 0.4 mg-kg DCZ. Aside from the bad quality of the plot, which makes it hard to read, this data is useless if not all recordings for all conditions are shown. Moreover, the authors mention repeated DCZ application over multiple days. However, they do not provide a precise protocol and it is not clear in which temporal scheme the different DCZ doses were administered.
The doses of DCZ used here were relatively high and behavioral effects became only prevalent (if one believes the author’s verbal statements) at doses above 0.1 mg/kg. To rule out side effects of DCZ on the animal’s state, control injections into a non-virus-injected animal would be important.
Finally, from the information I could access, it is not clear what happened with monkey JS. Why was this monkey not used or why is it not mentioned with any of the experiments? Why was monkey A directly subjected to immunohistochemistry experiments? More generally, it is not clear why not all three monkeys were used for the pharmacological manipulations with DCZ. This would have allowed for some more meaningful analysis including statistics. As it is now, the data does not support the claim of the authors that their method provides a reliable way to selectively and reversibly inactivate LC neurons.
The authors make strong claims that expression of hM4Di is selective and specific for NA neurons. However, this claim is difficult to judge based on the data provided here. There are multiple issues that need to be addressed:
- Fluorescence and cell morphology are not easily visible in fig. 3. The authors need to show pure fluorescence images without underlying morphology images at sufficient resolution to see neuronal structures. As the data appear now, there seems to be very granulated fluorescence, which does not see very specific. Moreover, they need to show a larger field of view at lower magnification to be able to judge whether expression is indeed confined to neurons of the LC.
- Immuno stainings in figure 4 should also be shown for regions outside LC to better judge expression in brain tissue around LC. Again, as for figure 3, also a higher magnification is required to better see neurons.
- The color choice in figure 5 is bad and it is very difficult to assign the different channels in the merged figure. A better combination would be green and magenta. In addition, to substantiate their claim that their injection completely covers the LC and is specific to NA neurons, they need to provide images of better quality and they need to quantify the overlap between TH-positive and hMDi-positive cells. This will give a metric of false-positive and missed cells. Finally, images of LC sections along the rostro-caudal axis should be shown to demonstrate complete coverage of the LC.
The schematic drawing of the injection device in fig. 2 is not very accurate. Since the authors lay a strong focus on their new injection method, the device should be shown in a more professional drawing with proper annotations.
Author Response
Reviewer 1
In the present study, the authors describe a method to chemogenetically control the activity of the Locus Coeruleus (LC) in macaques. They use stereotaxic injection of Lentiviruses encoding hM4Di-mCherry or -CFP under the control of a fragment of the tyrosine hydroxylase (TH) promoter. The advantage of this promoter is that it should only be active in cells synthetizing catecholamines, as this requires the hydroxylation of tyrosine via TH. The authors used three monkeys in the study, A, JS, and JF into whose LC’s they injected variants of the hM4Di-Lentiviruses. They asses expression with immunohistochemistry and evaluate chemogenetic silencing of the LC by rating vigilance and by measuring single unit activity in the LC. In principle, the study is interesting and might provide some insights into better ways of controlling LC in macaques. However, there are a number of problems with the manuscript.
First, the authors make a number of claims for which they do not provide any or only minimal data. Most problematically, they rely heavily on the analysis of vigilance. However, there is not a single data point showing any quantification of this parameter. Moreover, a description how vigilance was measured or scored is missing. If quantifying vigilance proves difficult, one way to inspect the activity of LC in a more quantitative way could be to measure pupil dilation.
-> We agree that quantifications were missing, but the effects of high doses of DCZ were clear enough: the animal spent most of the time with their eyes closed, without any movement, whereas it was constantly awake before the injection. We have clarified this in the new version of the manuscript (section 5, ‘functional validation’). Measuring pupil diameter would have required some kind of fixation training, and maintaining stable fixation for such a long time would probably be problematic. But we will consider it for future experiments.
Next, the authors deduce the effects of different concentrations of deschloroclozapine (DCZ) on LC activity by recording single units. Also, in this case, almost no data is shown. There is one exemplary histogram showing firing of a single unit after 0.4 mg-kg DCZ. Aside from the bad quality of the plot, which makes it hard to read, this data is useless if not all recordings for all conditions are shown. Moreover, the authors mention repeated DCZ application over multiple days. However, they do not provide a precise protocol and it is not clear in which temporal scheme the different DCZ doses were administered.
-> We have significantly improved the description of the effects of the treatments on the neurons, and generated a new figure, as well as a table summarizing the statistics of individual neurons.
The doses of DCZ used here were relatively high and behavioral effects became only prevalent (if one believes the author’s verbal statements) at doses above 0.1 mg/kg. To rule out side effects of DCZ on the animal’s state, control injections into a non-virus-injected animal would be important.
-> We agree with the reviewer that such control would have been informative, but the fact that the influence of the drug on neuronal activity increases with the dose and the strong similarity between the effects described here and those of clonidine make it unlikely that the influence of DCZ can only be accounted for by without its action on the DREADDs expressed in the LC. We have addressed that point in more details with Reviewer 2, in relation to the work of Upright & Baxter, 2020, and we have edited the discussion to address that issue directly.
Finally, from the information I could access, it is not clear what happened with monkey JS. Why was this monkey not used or why is it not mentioned with any of the experiments? Why was monkey A directly subjected to immunohistochemistry experiments? More generally, it is not clear why not all three monkeys were used for the pharmacological manipulations with DCZ. This would have allowed for some more meaningful analysis including statistics. As it is now, the data does not support the claim of the authors that their method provides a reliable way to selectively and reversibly inactivate LC neurons.
-> Monkey JS was used to validate the safety and the reliability of the surgical procedure, and it will be tested with DCZ in a specific behavioral task. The brain of monkey A was directly extracted for the sake of time, and at times when running experiments was complicated because of COVID. We believe that the reviewer has little idea of the constraints associated with monkey research, including the difficulty of accurately and safely accessing it for injection or neurophysiology. We hope that this paper would be useful to researchers interested in studying the LC in primates, and we are sorry that the reviewer acts as if there were too many labs performing such experiments already,
The authors make strong claims that expression of hM4Di is selective and specific for NA neurons. However, this claim is difficult to judge based on the data provided here. There are multiple issues that need to be addressed:
- Fluorescence and cell morphology are not easily visible in fig. 3. The authors need to show pure fluorescence images without underlying morphology images at sufficient resolution to see neuronal structures. As the data appear now, there seems to be very granulated fluorescence, which does not see very specific. Moreover, they need to show a larger field of view at lower magnification to be able to judge whether expression is indeed confined to neurons of the LC.
- Immuno stainings in figure 4 should also be shown for regions outside LC to better judge expression in brain tissue around LC. Again, as for figure 3, also a higher magnification is required to better see neurons.
- The color choice in figure 5 is bad and it is very difficult to assign the different channels in the merged figure. A better combination would be green and magenta. In addition, to substantiate their claim that their injection completely covers the LC and is specific to NA neurons, they need to provide images of better quality and they need to quantify the overlap between TH-positive and hMDi-positive cells. This will give a metric of false-positive and missed cells. Finally, images of LC sections along the rostro-caudal axis should be shown to demonstrate complete coverage of the LC.
-> We have worked on the figures, but for the sake of time, we cannot perform these additional analysis.
The schematic drawing of the injection device in fig. 2 is not very accurate. Since the authors lay a strong focus on their new injection method, the device should be shown in a more professional drawing with proper annotations.
-> There seems to have been a problem with the annotations, and we hope that adding them to the figure will make it more professional.
Reviewer 2 Report
In vivo functional manipulation of noradrenergic locus coeruleus neurons is of critical importance for understanding the role of the LC in diverse behaviors and achieving this goal in primates is important for translational research. Perez et al describe a lentiviral pharmacogenetics approach to express hM4Di DREADD for selective and reversible inactivation of LC noradrenergic neurons. Their results indicate that they have successfully expressed hM4Di in the LC, but some deficits in data reporting should be corrected in the manuscript to better highlight their achievement.
- While we realize that it is difficult to obtain negative control tissue when working with monkeys, the authors should at least show technical controls for staining in Figures 4 and 5 (i.e. unstained sections or sections with no primary antibody). In the current figures, LC-specific staining is weak and background high, making it difficult to evaluate the authors’ claim that “a majority of LC neurons expressed the hM4Di DREADD” (line 285). This weak staining is surprising, given that endogenous CFP was visible in at least a few cells after sectioning (Fig 3B). Controls are particularly important for the DAB stain in Figure 4B, because of the natural pigmented in the monkey LC (Fig 3A).
- The authors should discuss the possibility of non-specific effects of DCZ at higher doses (0.4 and 0.5 mg/kg). Nagai et al. (2020) used a maximum dose of 0.1 mg/kg in DREADD-expressing monkeys. It has also been shown that 0.3 mg/kg DCZ impaired performance of monkeys on a working memory task in the absence of DREADD expression (Upright et al., 2020). Given that the current study lacks DREADD-negative controls, is there concern that the decrease in vigilance observed at higher DCZ doses could be an off-target effect of DCZ?
- The authors should report data for unit activity recording sessions at all doses, not just a single example at 0.4 mg/kg DCZ (Fig 6). The non-LC unit recordings (p.8, line 326-327) could serve as a useful control showing non-responding cells after DCZ administration. Furthermore, the authors’ claim that “…3 out of 6 LC neurons displayed a significant decrease in activity” (p. 10, line 400) implies that statistical analysis was performed. These results should be reported.
- Scale bar units should be “µm” or “mm” and not “pixel”.
- Fig 6 is not referenced in the main text.
- Fig 2A: The letters listed in the caption are not present in the diagram to identify the various parts of the injecting system (e.g., “g” for guide-tube, “s” for spacer, “i” for injector, etc.).
Author Response
Reviewer 2
In vivo functional manipulation of noradrenergic locus coeruleus neurons is of critical importance for understanding the role of the LC in diverse behaviors and achieving this goal in primates is important for translational research. Perez et al describe a lentiviral pharmacogenetics approach to express hM4Di DREADD for selective and reversible inactivation of LC noradrenergic neurons. Their results indicate that they have successfully expressed hM4Di in the LC, but some deficits in data reporting should be corrected in the manuscript to better highlight their achievement.
-> We thank the reviewer for this encouraging and constructive evaluation of our work.
- While we realize that it is difficult to obtain negative control tissue when working with monkeys, the authors should at least show technical controls for staining in Figures 4 and 5 (i.e. unstained sections or sections with no primary antibody). In the current figures, LC-specific staining is weak and background high, making it difficult to evaluate the authors’ claim that “a majority of LC neurons expressed the hM4Di DREADD” (line 285). This weak staining is surprising, given that endogenous CFP was visible in at least a few cells after sectioning (Fig 3B). Controls are particularly important for the DAB stain in Figure 4B, because of the natural pigmented in the monkey LC (Fig 3A).
-> We are sorry, but we do not have these controls. The weakness of the signal is probably associated to the poor signal/ noise ratio of the immuno-fluorescence here. We agree that a more thorough quantification of the signal would help evaluate the extent to which individual LC neurons express the DREADD, but it would take several months. We have attenuated our claims to acknowledge for this limitation.
- The authors should discuss the possibility of non-specific effects of DCZ at higher doses (0.4 and 0.5 mg/kg). Nagai et al. (2020) used a maximum dose of 0.1 mg/kg in DREADD-expressing monkeys. It has also been shown that 0.3 mg/kg DCZ impaired performance of monkeys on a working memory task in the absence of DREADD expression (Upright et al., 2020). Given that the current study lacks DREADD-negative controls, is there concern that the decrease in vigilance observed at higher DCZ doses could be an off-target effect of DCZ?
-> We agree with the reviewer, and we have edited the discussion to include that point (3rd page of the discussion, 2nd paragraph). One more general issue raised here is that since Upright et al only observed an off-target effect in 2/4 monkeys, inter-individual variability might be another factor to take into account for off-target effects, and thus characterizing it properly would require either a larger number of animal or a within subject design, which is not possible anymore here, unfortunately. In addition, we believe that an interpretation in terms of ‘off target effects’ is not necessarily the most parsimonious given the dose-response relation on neuronal activity: the small decrease in LC activity observed for low (specific) doses should be sufficient to account for the behavioral effects observed at higher doses, when LC neurons also display a stronger decrease in activity.
- The authors should report data for unit activity recording sessions at all doses, not just a single example at 0.4 mg/kg DCZ (Fig 6). The non-LC unit recordings (p.8, line 326-327) could serve as a useful control showing non-responding cells after DCZ administration. Furthermore, the authors’ claim that “…3 out of 6 LC neurons displayed a significant decrease in activity” (p. 10, line 400) implies that statistical analysis was performed. These results should be reported.
-> We agree with the reviewer, and we have profoundly edited the description of neuronal effects. We have generated a new figure and added a table with individual neurons.
- Scale bar units should be “µm” or “mm” and not “pixel”.
-> Thank you for pointing that out, we have edited the figures
- Fig 6 is not referenced in the main text.
-> Thank you for pointing that out, we have added a reference in section 5 of the result section and in the discussion.
- Fig 2A: The letters listed in the caption are not present in the diagram to identify the various parts of the injecting system (e.g., “g” for guide-tube, “s” for spacer, “i” for injector, etc.).
-> Thank you for pointing that out, we have edited the figures. We believe that there was a problem with the files. Hopefully the new version will include the letters.
Reviewer 3 Report
The manuscript from Perez et al. entitled “Using Inhibitory DREADDs to Silence LC Neurons in Monkeys” develops a genetic and viral strategy to efficiently deliver a chemogenetic tool (here hM4Di) to the Locus coeruleus. Their LC targeting strategy comprise a meticulous
and well described surgery and stereotactic injection procedure that allows a targeted infusion of virus into the region of the LC while limiting transduction of non-LC neurons. Furthermore, they described a well-though through lentiviral approach that usually leads to a smaller diffusion sphere than the more commonly used AAVs, but here enable the authors to further spatially restrict the area of viral infection. And finally, they incorporate a genetic approach that utilizes a TH-promoter (2.5kb that will only fit in packaging capacity of a lentivirus) that further improves targeting into catecholaminergic neurons. The authors verified expression and distribution of hM4Di with sophisticated immunohistology against a high autofluorescence that likely arises through a restricted perfusion after the death of the animal. Afterwards, the authors demonstrate that administration of various concentration of Deschloroclozapin (DCZ), a high-potent agonist of the expressed designer receptor that is known to be functional in monkeys, activate hM4Di in LC neurons. They authors show via extracellular recordings of 3 units that low dose of DCZ reduces firing rate in LC neurons, while a higher dose silence all recorded units (3 – I believe)
Overall the work is the first step to an exciting set of experiments in monkeys that will allow to recapitulated some milestone experiments from rodent research that in the context of a monkey model will help to shine new light on how LC is is regulating higher order cognitive processing and attention. Congratulation!
There are couple of things, I would like to comment on.
- Why did the authors choose to use a rat TH promoter in a monkey model? Just because the construct was available or is the sequence homology between TH promoter from rats and monkey small? Maybe add a sentence or two about which part of the promoter was chosen and why – there are quite some different promoter versions floating around in the community.
Also, there is an mTH promoter available that do not show much specificity to catecholaminergic neurons (81% in the VTA/ SNc – Chan et al, 2017) and (unpublished findings) even less than that for LC neurons.
- There is long controversy in the rodent research on how strong hM4D can actually inhibit LC neurons. Potentially the strong basal auto-inhibition via a2 receptor, and therefore constant high levels of active beta/gamma G proteins, is limiting the inhibition effect of hM4Di. Nevertheless, studies such as McCall et al, 2015 demonstrate convincingly that chemogenetic tools can inhibit LC neurons. Yet, I would like to suggest that the authors should include more population data so that this manuscript becomes a clear milestone paper for chemogenetic inhibition in monkeys. Maybe plot the average firing rate of all 9 LC units before and after injection of DCZ (even though some of them are not responsive)?
Also my version of figure 6 in the manuscript has a poor quality.
Maybe also show a rate histograms on a 3x3 subplots for each unit? I would also be curious to see if tonic activity is decreased or number of phasic events are preferentially suppressed. Do you have recordings at similar length of LC neurons in non-infected monkeys?
- In figure 5, on the lateral/ventral site of the LC, I can see rather strong fluorescence. Is that location closer to the injection site? Could the author comment on why these neurons are more clearly exhibiting fluorescence compared to neurons 20pixel further medial. How does a non-infectd monkey brain slice from a previous slice preparation looks like under similar imaging conditions?
Maybe also change the unit in length from pixel to micrometer or millimeter?
Minor:
In figure 3 the LC appears to be brownish. Is that similar to the neuromelanin pigmentation often seen in the SNpc in primates? Is that a typical coloring for the LC in monkeys? Or could it be an effect of the viral expression?
Page 6 – line 245 à spelling panel
Figure 5B à increase contrast (even saturate pixles), in my version I find a difficult to see the LC (only the overlay in C shows some yellowish fluorescence)
Page 3 – line 147 – Siemens
Page 9 – line 334 – vertical
Author state that inhibition can last several hours (page 9 351/52), but it is not clear to me how long the recording actually lasted (page 8 312 – states a least an hour). Maybe give a range?
Despite the impressive results and demonstration that the chosen lentiviral approach resulted in the desired spatial targeting, I believe that integration of transgene into genomic DNA that occurs with lentiviral delivery should be discussed. Potentially genomic integration could lead to mosaic expression patterns or potentially hampered long-term activation via epigenetic silencing mechanisms. Maybe add a sentence or two in the discussion.
Furthermore, are there evidence in rodent literature of a downregulation of DREADDs during repeated, long-term activation? Also here, maybe show percent of inactivation of a unit on a specific experimental day along different experimental dates. Even though these will be different units it give some experimental evidence that inactivation remains possible over long periode of time i.e. plot percent of inactivation of a given unit versus date of experiment. Page 11 430 – 32 is bit vague.
Authors always write CFP-hM4Di – does that mean the CFP fusion is N-terminal? If not, authors should change to hM4Di-CFP …
Author Response
Reviewer 3:
The manuscript from Perez et al. entitled “Using Inhibitory DREADDs to Silence LC Neurons in Monkeys” develops a genetic and viral strategy to efficiently deliver a chemogenetic tool (here hM4Di) to the Locus coeruleus. Their LC targeting strategy comprise a meticulous
and well described surgery and stereotactic injection procedure that allows a targeted infusion of virus into the region of the LC while limiting transduction of non-LC neurons. Furthermore, they described a well-though through lentiviral approach that usually leads to a smaller diffusion sphere than the more commonly used AAVs, but here enable the authors to further spatially restrict the area of viral infection. And finally, they incorporate a genetic approach that utilizes a TH-promoter (2.5kb that will only fit in packaging capacity of a lentivirus) that further improves targeting into catecholaminergic neurons. The authors verified expression and distribution of hM4Di with sophisticated immunohistology against a high autofluorescence that likely arises through a restricted perfusion after the death of the animal. Afterwards, the authors demonstrate that administration of various concentration of Deschloroclozapin (DCZ), a high-potent agonist of the expressed designer receptor that is known to be functional in monkeys, activate hM4Di in LC neurons. They authors show via extracellular recordings of 3 units that low dose of DCZ reduces firing rate in LC neurons, while a higher dose silence all recorded units (3 – I believe)
Overall the work is the first step to an exciting set of experiments in monkeys that will allow to recapitulated some milestone experiments from rodent research that in the context of a monkey model will help to shine new light on how LC is is regulating higher order cognitive processing and attention. Congratulation!
-> We thank the reviewer for these positive and encouraging comments. They are really welcome.
There are couple of things, I would like to comment on.
- Why did the authors choose to use a rat TH promoter in a monkey model? Just because the construct was available or is the sequence homology between TH promoter from rats and monkey small? Maybe add a sentence or two about which part of the promoter was chosen and why – there are quite some different promoter versions floating around in the community. Also, there is an mTH promoter available that do not show much specificity to catecholaminergic neurons (81% in the VTA/ SNc – Chan et al, 2017) and (unpublished findings) even less than that for LC neurons.
-> We used a 2362 pb long fragment of the rat TH promoter to drive expression of the hM4Di transgene. This fragment correspond to the proximal TH promoter that includes the transcription start site. Such area of the promoter shows strong homology with Macaque and human promoter (>80%). In addition based on P. Ravassard unpublished data we have observed that such fragment is able to restrict expression of a reporter gene in human iPS derived Dopaminergic and Noradrenergic cells. We have added this information in the corresponding section of the article (2-Viral Construct/ lentiviral construct).
- There is long controversy in the rodent research on how strong hM4D can actually inhibit LC neurons. Potentially the strong basal auto-inhibition via a2 receptor, and therefore constant high levels of active beta/gamma G proteins, is limiting the inhibition effect of hM4Di. Nevertheless, studies such as McCall et al, 2015 demonstrate convincingly that chemogenetic tools can inhibit LC neurons. Yet, I would like to suggest that the authors should include more population data so that this manuscript becomes a clear milestone paper for chemogenetic inhibition in monkeys. Maybe plot the average firing rate of all 9 LC units before and after injection of DCZ (even though some of them are not responsive)?Also my version of figure 6 in the manuscript has a poor quality. Maybe also show a rate histograms on a 3x3 subplots for each unit?
-> We agree with the reviewer that the description of the neural data was not adequat. We have profoundly updated that section, and generated a figure with population data. We have also provided a table with statistical data for individual units.
I would also be curious to see if tonic activity is decreased or number of phasic events are preferentially suppressed. Do you have recordings at similar length of LC neurons in non-infected monkeys?
-> Tonic vs phasic is only a function of the window used for analysis: (ie looking at phasic events only means using a shorter window to count spikes, and as far as we know there is no accepted standard (or solid physiological evidence) for labeling rate changes as ‘tonic’ or ‘phasic’. Is 1 second tonic or phasic? how about 1 minute? etc etc. We believe that these terms are generally confusing, especially when mixed up with the model of Aston-Jones and Cohen on firing mode, which implies a change in coherence among neurones at least as much as a change in firing rate of individual units. So we try to avoid that terminology. That said, what could have been interesting, would have been to compare the effect of DCZ on ‘spontaneous’ vs stimulus-evoked LC activation, but we have not used formal events to trigger well controlled ‘phasic’ responses (typical activation/ inhibition responses, lasting a few hundred milliseconds). So we cannot address that issue here, but we this is at the very top of our list for the next study, in monkeys performing a behavioral task.
-> We do not have recording of similar length with DCZ injections in non-infected monkeys. But we hope to be able to obtain that data in future experiments, to address the issue of specific vs off-target effects, as pointed out by reviewer 2.
- In figure 5, on the lateral/ventral site of the LC, I can see rather strong fluorescence. Is that location closer to the injection site? Could the author comment on why these neurons are more clearly exhibiting fluorescence compared to neurons 20pixel further medial. How does a non-infectd monkey brain slice from a previous slice preparation looks like under similar imaging conditions?
-> This is a good question, but we do not know. It is probably not only about DREADDs since TH shows the same effect. And it is not there on all sections (see fig 4A). We have tried to address this issue (heterogeneity in staining) in the discussion, including a reference to the amazing work of Manger & coll.
Maybe also change the unit in length from pixel to micrometer or millimeter?
-> Definitely! Thank you for pointing that out.
Minor:
In figure 3 the LC appears to be brownish. Is that similar to the neuromelanin pigmentation often seen in the SNpc in primates? Is that a typical coloring for the LC in monkeys? Or could it be an effect of the viral expression?
-> this is typical, indeed, and indeed similar between SNc and LC (locus niger and locus coeruleus…)
Page 6 – line 245 à spelling panel
Figure 5B à increase contrast (even saturate pixles), in my version I find a difficult to see the LC (only the overlay in C shows some yellowish fluorescence)
-> ok, we have tried to improve also figure 4A.
Page 3 – line 147 – Siemens
Page 9 – line 334 – vertical
-> ok
Author state that inhibition can last several hours (page 9 351/52), but it is not clear to me how long the recording actually lasted (page 8 312 – states a least an hour). Maybe give a range?
-> Thank you for pointing that out. We have added that information in the text (section 5, on fonctionnal effects).
Despite the impressive results and demonstration that the chosen lentiviral approach resulted in the desired spatial targeting, I believe that integration of transgene into genomic DNA that occurs with lentiviral delivery should be discussed. Potentially genomic integration could lead to mosaic expression patterns or potentially hampered long-term activation via epigenetic silencing mechanisms. Maybe add a sentence or two in the discussion.
-> Thank you, we have edited the text accordingly:
Lentiviral vectors are integrative vectors. Integration although considered as random is in fact preferentially observed in active regions of the chromatin (Vranckx et al, 2016). In addition transduced cells often contain multiple integration sites that usually normalize the expression level between transduced cells that are all different in terms of chromosomal integration. It is true that epigenetic silencing could contribute to reduce long term expression and this is clearly observed when using for example a CMV promoter that is well known to be sensitive to methylation (Norrman et al, 2010). When using cell specific promoters the situation is different, since such promoters are active in the target cell type and therefore will not be epigenetically extinguished. Integration site could be responsible for the epigenetic extinction of the exogeneous promoter but this will occur in small number of transduced cells integration is different from one cell to the other.
Furthermore, are there evidence in rodent literature of a downregulation of DREADDs during repeated, long-term activation? Also here, maybe show percent of inactivation of a unit on a specific experimental day along different experimental dates. Even though these will be different units it give some experimental evidence that inactivation remains possible over long periode of time i.e. plot percent of inactivation of a given unit versus date of experiment. Page 11 430 – 32 is bit vague.
-> the experiments were conducted within less than 2 weeks, and it would be strongly confounded by the effect of doses of DCZ. Unfortunately, our data is probably too preliminary to address this issue, but we hope to be able to address it in the future.
Authors always write CFP-hM4Di – does that mean the CFP fusion is N-terminal? If not, authors should change to hM4Di-CFP …
-> The fusion of the CFP reporter gene was made a the C-terminal part of the hM4Di protein. Will have therefore used hM4Di-CFP in the text as suggested by the reviewer.
Round 2
Reviewer 1 Report
The authors have not seriously addressed most of my concerns. The only point that was significantly improved was the quantification and visualization of the LC electrophysiology. Otherwise, many statements in the manuscript are still not backed up by any data. They could have just toned down these statements or conclusions. Instead, they still make strong statements about vigilance without showing any kind of data or objective quantification. Moreover, the quality of the histology is still not suitable to substantiate their statements. No quantification is presented, nor negative controls for autofluorescence or staining specificity. The authors rather chose to judge the competence of the reviewer in their response. Be assured, the reviewer is well-aware of the constraints associated with monkey work. It would be simply sufficient to report these constraints and explain better the limitations to the general reader. I leave it to the editor of this journal to decide if they see the manuscript fit for publication.